# Transcriptome Profiling of Two *Camellia japonica* Cultivars with Different Heat Tolerance Reveals Heat Stress Response Mechanisms

**DOI:** 10.3390/plants13213089

**Published:** 2024-11-02

**Authors:** Yue Tan, Yinzhu Cao, Fenglian Mou, Bin Liu, Huafeng Wu, Shihui Zou, Lijiao Ai, Shunzhao Sui

**Affiliations:** 1Chongqing Key Laboratory of Germplasm Innovation and Utilization of Native Plants, Chongqing Landscape and Gardening Research Institute, Chongqing 400715, China; a974371416@163.com (Y.T.); zsh15220@email.swu.edu.cn (S.Z.); 2Chongqing Engineering Research Center for Floriculture, College of Horticulture and Landscape, Southwest University, Chongqing 401329, China; yinzhu202108@163.com (Y.C.); 18223017481@163.com (F.M.); 15705983137@163.com (B.L.); swuwhf@126.com (H.W.)

**Keywords:** *Camellia japonica*, heat stress, physiological index, transcriptome

## Abstract

Camellia (*Camellia japonica*) is a semi-shaded plant that is highly vulnerable to heat stress. To investigate the mechanisms underlying heat stress in *C. japonica*, two *C. japonica* cultivars, “Xiaotaohong” and “Zhuapolian”, which exhibit significant differences in heat tolerance, were selected from four common cultivars. The selection methods included phenotypic observations and physiological index detection, including relative electric conductivity (REC), malondialdehyde (MDA) content, superoxide dismutase (SOD) enzyme activity, relative water content (RWC), and chlorophyll content. RNA-seq analysis yielded 980 million reads and identified 68,455 differentially expressed genes (DEGs) in the two *C. japonica* cultivars during heat stress compared to the control samples. Totals of 12,565 and 16,046 DEGs were differentially expressed at 16 h and 32 h, respectively, in “Xiaotaohong” during heat stress. In “Zhuapolian”, 40,280 and 37,539 DEGs were found at 16 h and 32 h, respectively. KEGG enrichment analysis revealed that both cultivars were enriched in the “plant hormone signal transduction” and “circadian rhythm” pathways at two stages, indicating the critical role these pathways play in the heat stress response. The differences in the tolerance between the two cultivars are likely linked to pathways such as “plant hormone signal transduction”, “photosynthesis”, and “circadian rhythm”. Some members of heat shock proteins (HSPs) are associated with the heat stress response. It is speculated that transcription factor families contributing to the tolerance differences include AP2/ERF, C3H, bHLH, bZIP, and MYB-related with a small number of heat shock factors (HSFs) also induced by the stress. In conclusion, these results reveal the changes in the physiological indices and molecular networks of two *C. japonica* cultivars under heat stress. This study lays the foundation for the breeding of superior heat-resistant *C. japonica* cultivars and for further molecular research.

## 1. Introduction

The genus *Camellia* contains approximately 250 species that are widely distributed across tropical and subtropical regions [1]. These species provide value to the landscape, as well as to the food and pharmaceutical industries [2]. *C. japonica*, the most important ornamental species in *Camellia*, is a woody ornamental plant, with over 32,000 registered cultivars [3]. Its optimal growth temperature lies between 18 and 25 °C. Its growth halts above 30 °C, and when temperatures exceed 35 °C, heat damage such as leaf burn, failure in flower bud differentiation, or even plant death may occur [4]. These conditions significantly reduce the economic value of *C. japonica*. In recent years, global warming has intensified, leading to extreme high temperatures in some regions [5], posing a threat to the preservation of *C. japonica* germplasm resources [6], and severely hindering the development of the *C. japonica* industry.

When temperatures reach the upper limits of a plant’s adaptation range, its metabolism processes, growth, and development are adversely affected. Generally, plants are considered to be under heat stress when exposed to temperatures 10–15 °C above their optimal range [7]. Considering the various research purposes and requirements, different research methods have been considered for heat stress treatment. Currently, the most common methods include leaf chambers, plant chambers or greenhouses, field-based tents, radiant heaters, and naturally hot summer months [7]. Research on plant heat stress is mainly divided into three categories: climate warming, heat waves, and heat shock. The difference lies in the timescale of the stress treatment and the amplitude of temperature change to simulate the heat stress that plants receive under different climatic conditions [8]. In this study, heat shock conditions were simulated in a plant chamber to assess the impact of stress on *C. japonica.*

Heat stress disrupts plant cell homeostasis, affecting plant growth and development [9]. The ornamental value of *C. japonica* is mainly due to its flowers. Heat stress during the reproductive stage disrupts its carbohydrate metabolism and damages its reproductive functions, ultimately affecting seed development. The current research indicates that heat stress can impair plant cell membrane function, cause protein denaturation, damage nucleotides, and lead to the accumulation of reactive oxygen species (ROS) [10,11]. Heat stress increases the contents of malondialdehyde (MDA) and ROS and causes the peroxidation of membrane lipids, leading to an increase in the plasma membrane permeability, as well as increased permeability of the cell membranes to ions, small molecules, and water. Consequently, the ion balance is disrupted [12]. ROS are mainly eliminated by antioxidant enzymes, which plants activate to reduce cellular oxidation. However, when stress pressure exceeds a threshold, the activity of antioxidant enzymes is reduced [13]. Photosynthesis is highly sensitive to temperature, and changes in chlorophyll content can reflect plant stress levels. Notably, heat stress reduces the chlorophyll a and chlorophyll b contents [14]. In plants such as wheat (*Triticum aestivum*) [15], potato (*Solanum tuberosum*) [16], and rhododendron (*Rhododendron latoucheae*) [17], physiological responses to heat stress are typically measured by indices such as antioxidant enzyme activity, MDA content, and chlorophyll content.

RNA-seq offers an efficient and cost-effective method for studying non-model organisms with a sequencing technique. Through de novo analysis methods, transcript information can be obtained without the reference genome database, which can reveal the regulatory mechanisms of plant stress response, growth, and development [18]. Recently, transcriptome analysis has been used to study the molecular networks involved in heat stress responses in various plants. These studies have indicated that the heat shock factor–heat shock protein (HSF–HSP), Ca^2+^-calmodulin, reactive oxygen species, and plant hormone pathways are involved in the plant heat stress response. Among these pathways, the HSF–HSP pathway plays a crucial role [19,20,21].

Currently, the research on heat stress in *C. japonica* is still at the physiological stage [22,23], leaving many gaps in the molecular understanding of its response to heat stress. This study fills some of these gaps by offering valuable insights into the transcriptional mechanisms underlying heat tolerance in *C. japonica*, providing a reference for evaluating and improving heat tolerance in this species.

## 2. Results

### 2.1. Leaf Burn Severity and Physiological Changes Induced by Heat Stress

The heat stress treatment was administered when the adaptive culture was complete. In the heat stress treatment, the leaves of the four *C. japonica* cultivars all showed different degrees of burns. After 16 h of treatment, “Xiaotaohong”, “Shibaxueshi”, and “Qixinhong” showed noticeable leaf discoloration, exceeding 20%, while “Zhuapolian” had less severe burning. After 32 h, “Xiaotaohong” and “Shibaxueshi” had experienced substantial leaf burn, exceeding 40%, whereas “Zhuapolian” showed milder damage (Figure 1a,b). Leaves were collected at 0, 16, and 32 h for subsequent analysis.

After the treatment and collection of the leaves, five physiological indices were assessed. The MDA and relative electric conductivity increased over time, while the SOD enzyme activity initially rose and then fell. The chlorophyll content and relative water content decreased. These changes indicate worsening damage to the plant cell membranes, compromised membrane function, involvement of the intracellular antioxidant enzyme system in stress response, affected photosynthesis, and reduced relative water content due to heat stress. “Zhuapolian” had better membrane integrity during this treatment, because its REC and MDA content were significantly lower than the other cultivars. The SOD activity, RWC, and chlorophyll content of “Zhuapolian” remained relatively high at 16 h and 32 h, suggesting that the physiological process of “Zhuapolian” was relatively not impacted. In contrast, “Xiaotaohong” performed the worst. (Figure 1c–g). The SOD activity, RWC, and chlorophyll content were positively correlated with the tolerance, and the REC and MDA content were negatively correlated with the tolerance. The results the corresponding formula are shown in Table 1. Phenotypic observation and average membership function analysis revealed that “Zhuapolian” exhibited the strongest heat tolerance, whereas “Xiaotaohong” showed the weakest tolerance (Table 1).

### 2.2. Global Transcriptomic Changes

To investigate the differences in heat tolerance between “Zhuapolian” and “Xiaotaohong”, we analyzed both cultivars at the molecular level. After filtering and removing low-quality reads, we obtained an average of 7.89 Gb of clean reads per sample, with an average GC content of 45.32% (Appendix A; Appendix A). Splicing produced 335,764 transcripts with an average length of 880 bp and N50 and N90 values of 1399 bp and 363 bp, respectively. A total of 197,478 unigenes, also averaging 880 bp, were annotated across the KEGG, NR, Swiss-Prot, GO, Pfam, and TREMBL databases (Appendix A; Appendix A). In the NR database, the top species similar to *C. japonica* were *Camellia sinensis* (43.31%), *Camellia lanceoleosa* (32.86%), *Camellia sinensis* var. Sinensis (8.94%), *Daucus carota* subsp. Sativus (1.31%), and *Quercus suber* (0.76%) (Appendix A). Sample correlation and principal component analyses confirmed the effectiveness of the biological replicates, showing good correlation with coefficients above 0.85 and no outlier samples (Appendix A). To validate the RNA-seq data, we performed qRT-PCR on 10 DEGs in both *C. japonica* cultivars at 0, 16, and 32 h. The qRT-PCR results corroborate the RNA-seq data (Appendix A).

### 2.3. Time-Course RNA-Seq Analysis

A total of 68,455 DEGs were identified in the two cultivars at the three treatment times points. At 0 h, there were 27,357 DEGs, with 15,463 upregulated and 11,894 downregulated. At 16 h, the number of DEGs rose to 47,415, including 27,702 upregulated and 19,713 downregulated. At 32 h, there were 47,120 DEGs, with 27,798 upregulated and 19,322 downregulated (Figure 1a). An internal comparison showed that at 16 h, “Xiaotaohong” had 12,565 DEGs, including 7043 upregulated and 5522 downregulated. At 32 h, “Xiaotaohong” had 16,046 DEGs, with 8996 upregulated and 7050 downregulated (Figure 1b). In “Zhuapolian”, there were 40,280 DEGs at 16 h and 37,539 at 32 h. Of these, 22,148 and 21,393 were upregulated, while 18,132 and 16,146 were downregulated, respectively (Figure 1c).

KEGG enrichment analysis revealed that both cultivars were significantly enriched in the “plant hormone signal transduction” and “circadian rhythm” pathways, highlighting their crucial role in the heat stress response. Analysis at the same time points showed enrichment in the pathways related to “plant hormone signal transduction”, “photosynthesis”, “circadian rhythm” (Figure 2d; Appendix A). GO enrichment analysis indicated that differences between the cultivars were linked to the photosystem, the regulation of seedling development, and histone modification (Appendix A). These differences contribute to the varying tolerance levels of the two cultivars to heat stress.

K-means cluster analysis categorized the DEGs into six clusters: upregulated in both cultivars (Cluster 1); downregulated in both (Cluster 2); first downregulated and then upregulated in “Zhuapolian” (Cluster 3); first upregulated and then downregulated in “Zhuapolian” (Cluster 4); first upregulated and then downregulated in “Xiaotaohong”, and downregulated in “Zhuapolian” (Cluster 5); and upregulated in “Xiaotaohong” (Cluster 6) (Figure 2e). Cluster 1, which was upregulated in both cultivars, showed enrichment in the KEGG pathways related to plant–pathogen interactions, spliceosome function, and protein processing in the endoplasmic reticulum.

Regarding plant hormone signaling, induced genes included auxin-responsive proteins, gibberellin receptors, and histidine kinases. In photosynthesis, genes encoding oxygen-evolving enhancer proteins, ferredoxin, and chlorophyll a-b binding proteins were upregulated. For circadian rhythm, genes such as phytochrome A and phytochrome C were induced. These genes likely play a significant role in the heat stress response and may explain the differences in heat tolerance between the two *C. japonica* cultivars (Appendix A).

### 2.4. Screening of HSPs

Under heat stress, the two cultivars of *C. japonica* exhibited different responses in *HSPs*. Some *HSPs* were induced and played roles in the plants’ heat stress response networks. By comparing the treatment processes of the two cultivars, it was observed that the degree of change in some heat shock proteins varied at different treatment times. For example, “Cluster-11922.3” and “Cluster-70892.0” from the *HSP90* family, “Cluster-52031.14”, “Cluster-58070.31”, “Cluster-58070.36”, “Cluster-58070.37”, “Cluster-58070.53”, and other members of the *HSP70* family, and “Cluster-28081.15” and “Cluster-5384.11” from the *HSP20* family all showed differences (Figure 3). KEGG annotations identified the *HSP90s* as being involved in the pathways “ko04141” and “ko04626”, corresponding to “protein processing in the endoplasmic reticulum” and “plant–pathogen interactions”. The *HSP70s* were associated with the pathways “ko03040”, “ko03060”, “ko04141”, and “ko04144”, which included “spliceosome”, “protein export”, “protein processing in the endoplasmic reticulum”, and “endocytosis.” The *HSP20s* were mainly linked to the “protein processing in the endoplasmic reticulum” pathway (ko04141). The *HSPs* are shown in Appendix A. These pathways are illustrated in Figure 4, Figure 5, Figure 6, Figure 7 and Figure 8.

### 2.5. Screening of TFs

Comprehensive analysis of the transcription factor families revealed the presence of many TFs in the C3H, AP2/ERF-ERF, MYB-related, bHLH, bZIP, and SET families. At 16 h of treatment, the relative TFs in “Xiaotaohong” were primarily from the AP2/ERF-ERF, bHLH, GARP-G2-like, MYB-related, MYB, NAC, bZIP, WRKY, HSF, and other families. At 32 h, the predominant TF families in “Xiaotaohong” were AP2/ERF-ERF, MYB-related, bHLH, GARP-G2-like, NAC, WRKY, TCP, and others. In “Zhuapolian”, the relative TFs at 16 h were mainly from the AP2/ERF-ERF, C3H, bHLH, bZIP, MYB-related, C2H2, SET, FAR1, MYB, and other families. At 32 h, the TFs in “Zhuapolian” were predominantly from the SET, FAR1, C3H, PHD, SNF2, AP2/ERF-ERF, and other families. While both cultivars had the same TF families at different stages, the numbers involved varied. They also contained unique TF families. Differences in the expressions of TF family genes resulted in varied responses between the two cultivars. It was speculated that the TFs contributing to the difference in the tolerance were mainly AP2/ERF, C3H, bHLH, bZIP, MYB-related, C2H2, SET, NAC, GARP-G2-like, and FAR1 (Appendix A). The numbers of induced TFs, such as HSFB2A, HSFB2B, and HSF30, were small (Appendix A). The differing expression levels of HSF in the two cultivars might explain the variations in their tolerance.

## 3. Materials and Methods

### 3.1. Plant Materials and Treatment

The four *C. japonica* cultivars used in this experiment (“Xiaotaohong”, “Shibaxueshi”, “Qixinhong”, and “Zhuapolian”) were collected from Longyan city, Fujian province, China. Three-year-old *C. japonica* plants were potted in a greenhouse at Southwest University in acidic peat soil.

At the beginning of the experiment, the four cultivars of *C. japonica* plants were placed in an artificial climate chamber with the following parameters: 25 °C light/20 °C dark, a 16 h light/8 h dark photoperiod, light intensity of 14,000 lx, and 75% humidity. They were cultured for 7 d. For heat stress testing, the temperature was adjusted to 42 °C light/37 °C dark, while the other conditions remained constant. The leaves were collected at 0, 16, and 32 h for physiological and transcriptome analysis, and images were captured using a camera.

### 3.2. Physiological Analysis

A conductivity meter (DDS-11A) was used to determine the relative electrical conductivity (REC) [24]. Fresh leaves were cut into small pieces, placed in tubes containing 20 mL of deionized water, and soaked for 3 h with gentle stirring. Subsequently, the conductivity was measured using an electrical conductivity meter to obtain the initial conductivity reading (R1). Tubes were then placed in a water bath and heated for 15 min to destroy plant tissue. After cooling, the sample conductivity was measured to determine its boiling conductivity (R2). The relative conductivity of the leaves was calculated as (R1/R2) × 100%.

The malondialdehyde (MDA) content was determined using the thiobarbituric acid method [25]. Fresh leaves (0.5 g) were extracted using 5% trichloroacetic acid (5 mL) and centrifuged at 5000× *g* for 20 min. Next, 2 mL of 0.67% (*w*/*v*) thiobarbituric acid was added to the 2 mL supernatant in a centrifuge tube. The mixture was heated for 30 min at 100 °C with shaking, and then cooled and centrifuged at 5000× *g* for 10 min. The absorbance values at 450, 532, and 600 nm were measured.

The superoxide dismutase (SOD) activity was detected using the nitro blue tetrazolium (NBT) method [26]. Leaves were ground into a powder in liquid nitrogen, and 200 mg of the ground leaves was homogenized in 1.6 mL of cold phosphate-buffered saline (PBS) (50 mM, pH 7.8). The mixture was centrifuged at 4 °C for 15 min at 10,000× *g*. The supernatants were retained for an antioxidant enzymatic assay, and the absorbance was measured at 550 nm.

The relative water content was determined using the mass method [27]. Leaves were removed from each treatment, and their fresh weights (FWs) were recorded immediately. After the leaves were immersed overnight in deionized water at 4 °C, the rehydrated weight (RW) was determined. The dry weight (DW) was then measured after drying the leaves in an oven at 80 °C for 48 h. The relative water content (RWC) (%) was calculated as follows: RWC (%) = (FW − DW)/(RW − DW) × 100.

The chlorophyll content was measured using the acetone–ethanol mixing method [28]. Fresh leaves were cut into pieces and immersed in a mixture of 20 mL acetone and anhydrous ethanol in a 1:1 ratio. The mixture was kept at 25 °C, and the leaves were soaked for 24 h until they had completely whitened. A volume of 200 μL of the supernatant was placed in a 96-well microtiter plate. The absorbance of the supernatant at 633 nm and 645 nm was measured using a Varioskan Flash Spectral Scanning Multimode Reader. The relative total chlorophyll content was calculated as (8.02 × A633 + 20.21 × A645) · 1000 V/W mg fresh weight.

The resulting physiological data were plotted using GraphPad software (version 9.5.1) and analyzed for statistical significance. The method of statistical analysis was one-way analysis of variance. The uppercase letters in the figure indicate significant differences among the same cultivars at different treatment times, while the lowercase letters indicate significant differences among the four *C. japonica* cultivars in the group at the same time.

To assess the heat tolerance, the membership function method was used. For indices positively correlated with the tolerance, the membership function value U (Xsy) was calculated as U (Xsy) = (Xsy − Xymin)/(Xymax − Xymin), where U (Xsy) is the membership function value, Xsy is the measured value of the plant under heat stress regarding the y-index, Xymin is the minimum value of all tested plants under heat stress regarding the y-index, and Xymax is the maximum value of the tested plants under heat stress regarding the y-index. For indices negatively correlated with the tolerance, the calculation was U (Xsy) = 1 − (Xsy − Xymin)/(Xymax − Xymin), the average value of the U (Xsy) results of the three time points was calculated as the membership function value of each index. The average of the membership function value of the five indices indicates the tolerance of each cultivar, with larger values representing better tolerance [29].

### 3.3. Transcriptomics

cDNA libraries were sequenced on an Illumina sequencing platform by Metware Biotechnology Co., Ltd. (Wuhan, China). Transcriptome assembly of clean reads was performed using Trinity (version 0.23.2). The assembled transcripts were clustered, and their redundancies were removed using Corset (version 1.09; https://github.com/trinityrnaseq/trinityrnaseq, accessed on 2 March 2024)). TransDecoder (version 5.3.0) was used to perform CDS prediction of the Trinity-assembled transcripts to obtain the corresponding amino acid sequences (https://github.com/TransDecoder/, access on 17 March 2024). After their de-redundancy, the transcript sequences were aligned with the KEGG, NR, Swiss-Prot, GO, COG/KOG, and TREMBL databases using DIAMOND software (version 2.0.9), and the amino acid sequences were aligned with the Pfam database using HMMER software (version 3.2) to obtain annotation information of the transcripts from these seven major databases. The expression levels of the transcripts were calculated using RSEM software (version 1.3.1), and then the FPKM of each transcript was calculated according to the transcript length. FPKM is currently the most commonly used method to estimate gene expression levels. Differentially expressed genes (DEGs) were analyzed using the DESeq2 software package in R (version 1.22.2), and then subjected to the Benjamini–Hochberg method for multiple hypothesis testing (i.e., |log_2_FC| ≥ 1, false discovery rate (FDR) < 0.05). Heat maps were constructed using the R package “pheatmap” (version 1.0.12) and TBtools software (version 2.030). Venn diagrams were generated using Venndiagram (version 1.6.20). Transcription factor predictions were made using iTAK (version 1.7a http://itak.feilab.net/cgi-bin/itak/index.cgi, access on 2 April 2024). Enrichment analysis was performed based on the hypergeometric test. For KEGG, the hypergeometric distribution test was performed based on the pathway; for GO, it was performed based on the GO term. The scale() function of the R language was employed to standardize the FPKM of the union of all DEGs, and Kmeans cluster analysis was conducted. The KEGG pathway figures were download from the KEGG official website (https://www.kegg.jp, access on 5 May 2024).

### 3.4. Quantitative Real-Time PCR (qRT-PCR)

To validate the RNA-seq data, we analyzed the expressions of 10 DEGs via qRT-PCR, including 7 involved in stress response and 2 related to circadian rhythm and photosynthesis. qRT-PCR was conducted using the 2×TSINGKE^®^ Master qPCR Mix kit (SYBR Green I with UDG) (TSE203, Beijing Tsingke Biotechnology Co., Ltd., Beijing, China) on the CFX96 Real-Time PCR Detection System (Bio-Rad, Hercules, California, USA) with the following conditions: 95 °C for 2 min, followed by 40 cycles of 95 °C for 15 s and 60 °C for 30 s, with a melt curve analysis from 65 to 95 °C, increasing 0.5 °C every 5 s. Three biological replicates were performed for each treatment, and each replicate contained three technical replicates. The 2^−ΔCT^ method was used to analyze the gene expressions [30]. The ubiquitin domain protein gene UBTD1 served as a reference gene, and the primers used for qRT-PCR are listed (Appendix A).

## 4. Discussion

The increase in greenhouse gases in the atmosphere has led to a continuous increase in the global average temperature. By 2100, the global average temperature is expected to rise by 0.3–4.8 °C [31]. Therefore, studying plant heat stress is an important future research direction. While previous transcriptome studies of *C. japonica* have focused on flower color and cold resistance, this study addresses the molecular aspects of heat stress in this species [32,33].

Plants undergo certain phenotypic changes under heat stress, which usually manifest as leaf discoloration, wilting, falling, and flower withering. These visible changes often serve as indicators of heat stress, making phenotypic observation a crucial first step in stress research. Under heat stress, the cell membrane structure is damaged, the cell membrane permeability changes, and internal substances leaks. To a certain extent, the conductivity reflects the integrity of the cell membrane [34]. Additionally, the MDA content can effectively reflect the degree of lipid peroxidation in plant cells [35,36,37]. The different membrane damage conditions among the four cultivars may be related to the different molecular response networks. Variations in membrane damage among the different cultivars may reflect distinct molecular response networks, with “Xiaotaohong” and “Zhuapolian” displaying different membrane protection mechanisms. Under heat stress, ROS accumulate in plant cells, and the internal structure of plant cells is damaged [38,39]. When plants are under heat stress, they activate antioxidant mechanisms to maintain the redox balance in cells [40]. In this study, the SOD enzyme activity increased in all four *C. japonica* cultivars early on, indicating a rapid response. However, the SOD activity declined later in “Xiaotaohong” and “Shibaxueshi”, suggesting that the stress exceeded their tolerance and the antioxidant systems began to fail. The relative water content is closely related to the metabolic activities of plant growth and development. A relatively stable water content is conducive to maintaining homeostasis in plant cells. Heat stress will cause water loss in plant cells, leading to wilting and death [41,42]. Despite starting with a slightly lower RWC, “Xiaotaohong” effectively maintained its water content and cell metabolic balance under heat stress, resulting in less damage compared to the other cultivars. Photosynthesis is crucial for plant life, with heat stress being considered one of the most sensitive environmental factors affecting plant photosynthesis [43]. A study indicated that plants with a strong tolerance to heat stress have a slower rate of decline in photosynthetic pigment content under heat stress [44]. Our findings, based on physiological indices such as REC, MDA content, SOD activity, RWC, and chlorophyll content, offer valuable insights into the heat tolerance of *C. japonica*.

The KEGG enrichment analysis revealed that the “plant hormone signal transduction”, “photosynthesis”, “circadian rhythm” pathways played crucial roles in the heat stress response. Plant hormone signal transduction is a biological mechanism through which plants adapt to environmental changes [45]. It plays an important role in plant morphology and the maintenance of physiological functions. Additionally, it has been shown to play an important role in the response to various abiotic stresses, such as drought, salinity, temperature, and waterlogging. Abscisic acid (ABA), auxin, brassinosteroid, cytokinin, and salicylic acid are crucial in plant responses to heat stress [46,47]. Photosynthesis is an important biological process that maintains the normal physiological functions in plants. The effect of heat stress on photosynthesis is mainly reflected in the inhibition of chlorophyll synthesis, which may be caused by the destruction of several enzymes in chlorophyll biosynthesis by heat stress [48], such as protochlorophyllide reductase and magnesium chelatase subunits [47]. Heat stress can also damage photosystems I and II (PSI and PSII). The water oxidation complex (WOC), PSII reaction center, and light-harvesting complex are impaired under heat stress, resulting in the obstruction of photosynthesis [49]. Circadian rhythm, which is an endogenous biological cycle, significantly influences plant metabolism, physiology, and development, and is essential for plant adaptability. Plant circadian rhythm is related to the accumulation and removal of ROS, indicating that it also plays an important role in heat stress [50]. In the future, it will be necessary to further investigate these pathways and identify the key molecules related to heat stress.

HSPs are vital for plant response to heat stress, playing roles in maintaining cell membrane stability and protein repair [51]. In our study, the analysis of DEGs showed that some HSPs in both cultivars had significantly increased expression early in the stress response. Heat shock protein 90 (HSP90) is a highly conserved and essential molecular chaperone involved in the maturation and activation of signaling proteins in eukaryotes [52]. HSP90 and its co-chaperones are involved in abiotic stress responses. Specifically, HSP90 and HSP70 negatively regulate the HSFs required for heat stress response, including the activation of a series of heat stress genes and multi-gene families encoding molecular chaperones [53,54]. HSP70 belongs to one of the most conserved protein families. As an important molecular chaperone, HSP70 functions in an ATP-dependent manner. Many proteins require the assistance of HSP70 to correctly fold and assemble into mature or native states, and its role also includes degrading damaged proteins. When plant cells are under heat stress, HSP70 and sHSPs are transferred to the cytoplasmic membrane and tonoplast to form peripheral proteins, which interact with cell membrane proteins and rebuild membrane fluidity [55]. HSP20s, also called small heat shock proteins (sHSPs)—are the most prevalent and abundant proteins in plant HSPs. Many HSP20s can form oligomers with high molecular weight and are involved in maintaining the stability of proteins, thus playing a vital role in the formation of plant acquired thermotolerance. HSP20s bind target proteins through conformational changes to prevent misfolding and irreversible protein aggregation [56]. In consideration of the function of these HSPs, their different expression levels in the two cultivars resulted in different heat tolerance. Therefore, research on these HSPs may be the key to screening heat-resistant plants and improving the heat tolerance of plants through molecular methods in the future.

AP2/ERF is one of the largest transcription factor families. It is widely involved in plant growth and development, secondary metabolism, stress effects, and other biological processes [57]. Analysis of the AP2/ERF family in Chinese cabbage (*Brassica rapa* ssp. pekinensis) revealed that eight of its members are involved in the temperature stress response [58]. C3H is a specific transcription factor containing a typical motif of three cysteine residues and one histidine residue. Analysis of the C3H family in potatoes revealed that many C3H members are expressed under heat stress, indicating that they play an important role in the potato response to heat stress [59]. bHLH, bZIP, and MYB-related genes have also been confirmed to play roles in the abiotic stress response of different plants [60,61]. Although only a small number of HSFs are involved in the heat stress response of *C. japonica*, they may play an important role in the regulatory network. HSFs are central to the heat stress response [62]. Activated HSFs bind to heat shock elements in the promoter regions of heat shock genes, regulating their expression [20]. In our study, these TFs were differentially induced in the two cultivars, resulting in the different heat tolerance, indicating their essential function in the heat stress response of *C. japonica*. In the future, research on the regulation function of these TFs will help us explore the heat stress response of *C. japonica*.

In this study, only five physiological indices were evaluated, and several other indices, including biomass, respiration rate, and peroxidase activity, were not discussed. Research on the physiological index is necessary to acquire a comprehensive understanding of *C. japonica* under heat stress. A certain number of HSPs and TFs have been screened, understanding the function and regulation of these HSPs and TFs will advance the study of heat stress response in *C. japonica.*

## 5. Conclusions

In this study, we used phenotype observation and physiological indices to compare the heat tolerance of four *C. japonica* cultivars, and we screened two cultivars with significant differences in their tolerance, “Xiaotaohong” and “Zhuapolian”. Through RNA-seq, we found that phytohormone signal transduction, circadian rhythm, photosynthesis, and other pathways play important roles in the heat tolerance of *C. japonica*, and we speculate that this is the main reason for the difference in tolerance between the two cultivars. The transcriptome data identified key heat shock proteins, including members of three families, HSP90, HSP70, and HSP20, and dozens of TFs related to the heat stress response. Future research on the key molecules in the response networks needs to be conducted.

## Figures and Tables

**Figure 1 plants-13-03089-f001:**
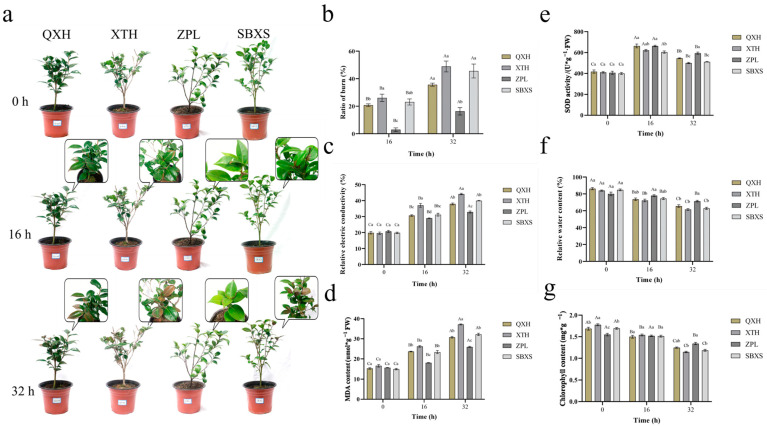
Phenotypic observation and results of the physiological index test. (**a**) Leaf burn in four *C. japonica* cultivars. (**b**) Ratio of leaf burn. (**c**) Relative electric conductivity. (**d**) MDA content. (**e**) SOD activity. (**f**) Relative water content (RWC). (**g**) Chlorophyll content. QXH: “Qixinhong”; XTH: “Xiaotaohong”; ZPL: “Zhuapolian”; SBXS: “Shibaxueshi”. The uppercase letters in the figure indicate significant differences among the same cultivars at different treatment times, while the lowercase letters indicate significant differences among the four C. japonica cultivars in the group at the same time.

**Figure 2 plants-13-03089-f002:**
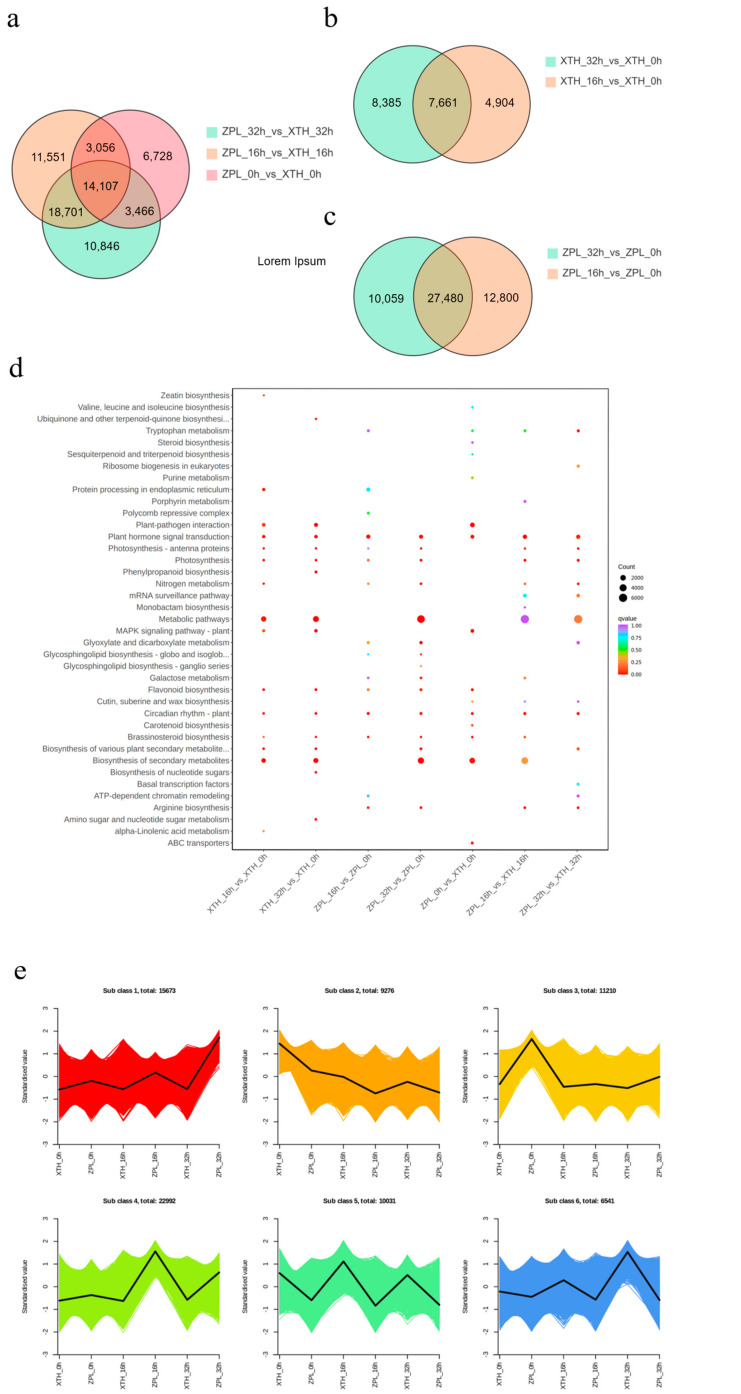
Comparative analysis of DEGs among samples, and KEGG enrichment analysis and K-means analysis of DEGs. (**a**) DEGs of two *C. japonica* species during the same period. (**b**) DEGs of “XTH”. (**c**) DEGs of “ZPL”. (**d**) KEGG enrichment of DEGs. (**e**) K-means clustering analysis of all DEGs. QXH: “Qixinhong”; XTH: “Xiaotaohong”; ZPL: “Zhuapolian”; SBXS: “Shibaxueshi”.

**Figure 3 plants-13-03089-f003:**
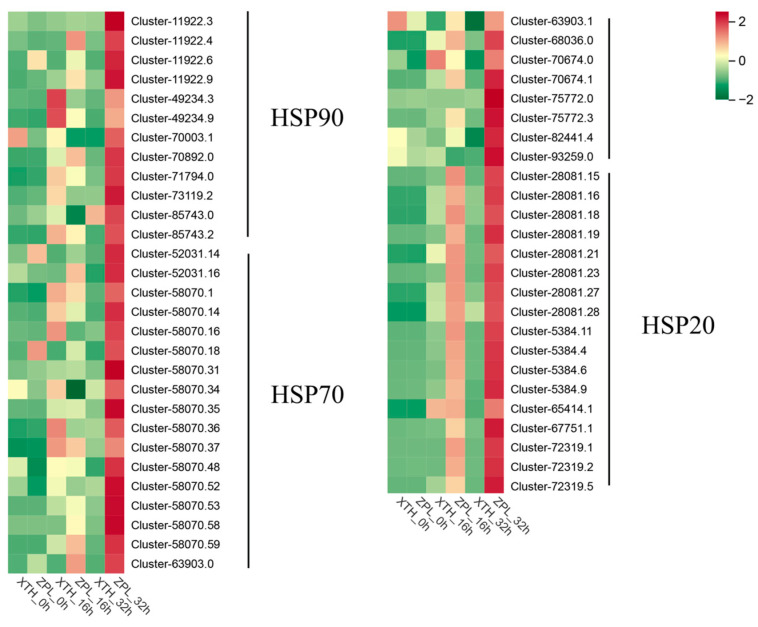
Heat map of related *HSPs*. Note: The *HSPs* were selected from Cluster 1. The colors of the blocks transition from green to yellow to red, indicating an increase in the degree of expression.

**Figure 4 plants-13-03089-f004:**
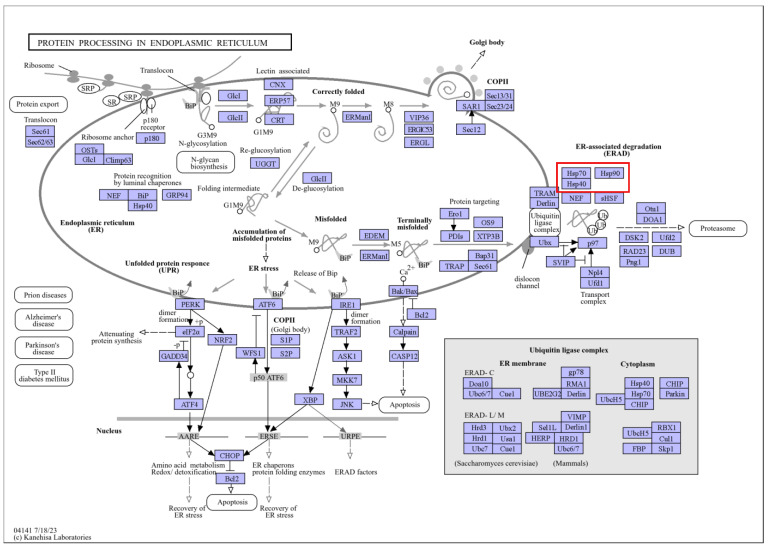
“Protein processing in the endoplasmic reticulum” pathways. Note: The red box marks the location of the HSP family.

**Figure 5 plants-13-03089-f005:**
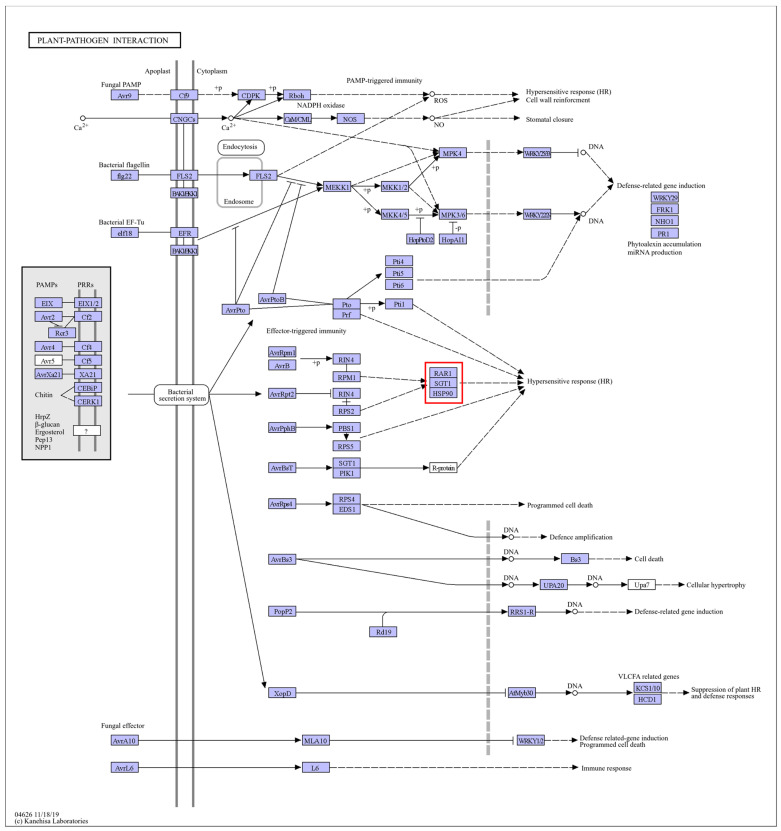
“Plant–pathogen interaction” pathways. Note: The red box marks the location of the HSP family.

**Figure 6 plants-13-03089-f006:**
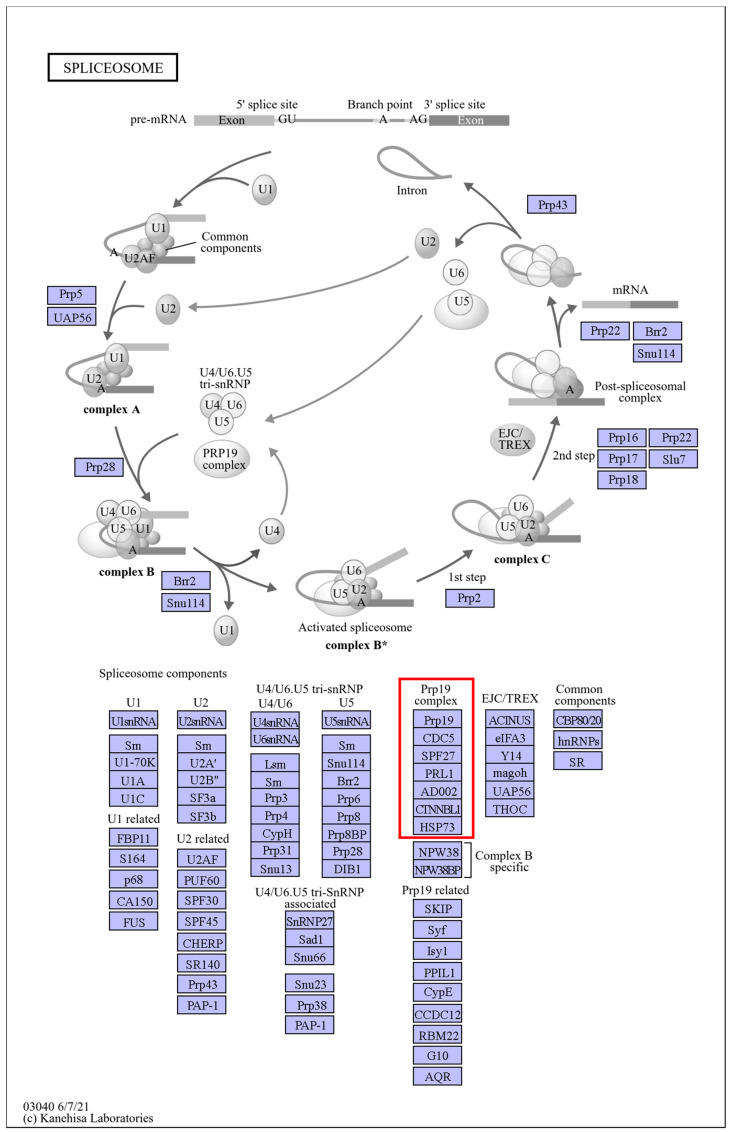
“Spliceosome” pathways. Note: The red box marks the location of the HSP family.

**Figure 7 plants-13-03089-f007:**
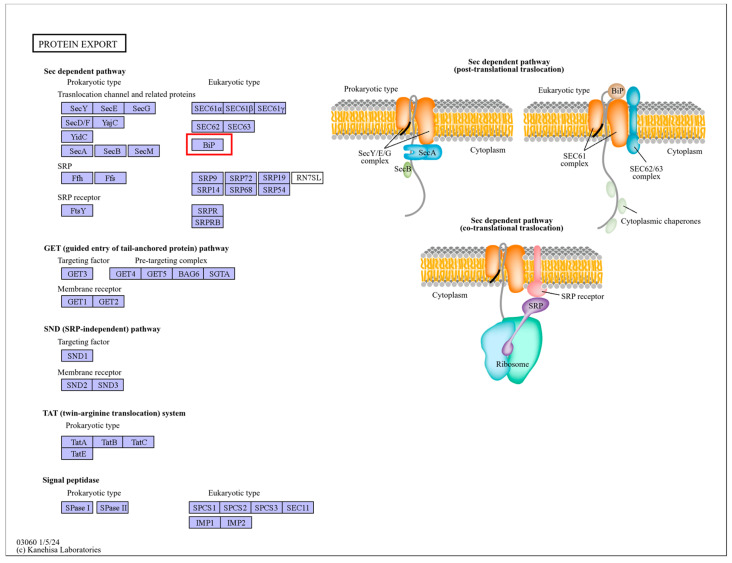
“Protein export” pathways. Note: The red box marks the location of the HSP family.

**Figure 8 plants-13-03089-f008:**
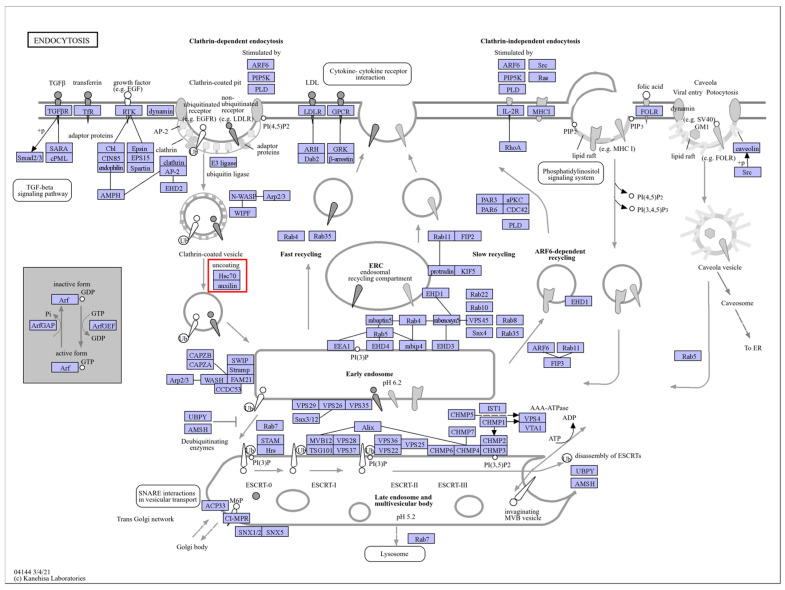
“Endocytosis” pathways. Note: The red box marks the location of the HSP family.

**Table 1 plants-13-03089-t001:** Membership function value and rank of heat tolerance of four *C. japonica* cultivars.

Cultivar	REC	MDA Content	SOD Activity	RWC	Chlorophyll Content	Average	RANK
“Qixinhong”	0.55	0.62	0.56	0.54	0.44	0.54	2
“Xiaotaohong”	0.43	0.47	0.45	0.45	0.53	0.47	4
“Zhuapolian”	0.66	0.76	0.61	0.59	0.52	0.63	1
“Shibaxueshi”	0.55	0.6	0.44	0.5	0.5	0.52	3

Note: The averages were calculated from the membership function values of the five physiological indices; the ranking is based on the average value.

## Data Availability

The data have been deposited to the National Center for Biotechnology Information (NCBI) under accession number PRJNA1158219.

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
