# Peer review of "Transcriptome Profiling of Two *Camellia japonica* Cultivars with Different Heat Tolerance Reveals Heat Stress Response Mechanisms"

_plants, 2024, doi:10.3390/plants13213089_

Round 1
Reviewer 1 Report
Comments and Suggestions for Authors
Overview:
The manuscript “Transcriptome profiling of two Camellia japonica varieties with different resistance reveals heat stress response mechanisms” investigated the mechanisms underlying heat stress in two C. japonica varieties, ‘Xiaotaohong’ and ‘Zhuapolian,’ which exhibit significant differences in heat resistance, selected from four common varieties through phenotypic observations and physiological index detection, including relative electric conductivity (REC), malondialdehyde (MDA) content, superoxide dismutase (SOD) enzyme activity and chlorophyll content. RNAseq analysis was then performed in order to offer valuable insights into the transcriptional mechanisms underlying heat stress resistance in C. japonica. I found the approach and the experimental design appropriate in order to reach the declared aim of the work, offering a comprehensive understanding of the heat stress resistance mechanisms. Overall, the current study is on a topic of relevance and general interest to the readers of the journal. However some minor revisions and concerns need to be addressed.
Minor comments
In my opinion, "heat tolerance" should be preferred over "resistance", that is a term generally used for defining genetic resistance against plant pathogens. Furthermore, resistance generally results in total absence of symptoms.
Materials and methods
· Section 2.2: the description of the GraphPad plot for physiological data should be reported in the results section, when figure is cited and described.
· Where the results of the membership function calculation are reported? No description of the measurements and parameters used in the U (XsY) formula is reported. I guess that the membership function somehow refers to the samples ranking in table 1, but it’s not clear. Please better detail this analysis and results.
· Section 2.3: Methods for GO and KEGG enrichment analysis of DEGs need to be reported. Please detail.
Results
· Figure 1: the quality and the resolution of the figure is low, labels and data are not readable. Please improve the quality of the figure and consider to split figure 1 in two different figures.
· Table 1: Better detail the table caption with parameter acronymous explanation. What “average” and “rank” columns refers to?
· Section 3.2: The percentage of annotated genes is a bit low (Figure S4, 73.96% unigenes Annotated in at least one Database). This is probably due to the fact that NT database of NCBI was not considered for unigenes annotation. I suggest to use also this database in order to increase the percentage of unigenes Annotated in at least one Database.
· Section 3.3: There is no information on how K-means cluster analysis for DEGs categorization was performed and obtained. Please describe it in method section
· Please provide data for clusters KEGG enrichment. No figures or tables report pathways and results you describe at the end of section 3.3.
· I suggest to provide a supplementary file reporting information of HSPs described and reported in figure 3 (similar to supplementary table 5 fot TFs).
Discussion
Discussion in some points need to be better linked with the obtained results. The literature cited is informative and the mechanisms of stress response are well described and discussed, but these information need to be related with the results you obtained. The question is: are the results obtained by this experiment in line with the mechanisms already known and described?
Author Response
Comments 1: In my opinion, "heat tolerance" should be preferred over "resistance", that is a term generally used for defining genetic resistance against plant pathogens. Furthermore, resistance generally results in total absence of symptoms.
Response: Thanks for your advice. After consideration, tolerance is more relevant to my research, and I have replaced resistance with tolerance in the manuscript.
Comments 2: Section 2.2: The description of the GraphPad plot for physiological data should be reported in the results section, when figure is cited and described.
Response: The description has been added in section 2.2.
Comments 3: Section 2.2: Where the results of the membership function calculation are reported? No description of the measurements and parameters used in the U (XsY) formula is reported. I guess that the membership function somehow refers to the samples ranking in table 1, but it’s not clear. Please better detail this analysis and results.
Response: Detailed description has been added, and the computational procedure is resubmitted in Non-published Material Table 1. Thanks for your correction.
Comments 4: Section 2.3: Methods for GO and KEGG enrichment analysis of DEGs need to be reported. Please detail.
Response: The methods for enrichment has been added.
Comments 5: Figure 1: the quality and the resolution of the figure is low, labels and data are not readable. Please improve the quality of the figure and consider to split figure 1 in two different figures.
Response: Thanks for your advice. I think it is more intuitive to put the information of this part together. Format of the picture is SVG, which can be viewed clearly after magnifying.
Comments 6: Table 1: Better detail the table caption with parameter acronymous explanation. What “average” and “rank” columns refers to?
Response: Note has been added in Table 1.
Comments 7: Section 3.2: The percentage of annotated genes is a bit low (Figure S4, 73.96% unigenes Annotated in at least one Database). This is probably due to the fact that NT database of NCBI was not considered for unigenes annotation. I suggest to use also this database in order to increase the percentage of unigenes Annotated in at least one Database.
Response: Thanks for your advice. We didn't notice the problem at first, future studies will address this.
Comments 8: Section 3.3: There is no information on how K-means cluster analysis for DEGs categorization was performed and obtained. Please describe it in method section.
Response: The method has been added in section 2.3.
Comments 9: Please provide data for clusters KEGG enrichment. No figures or tables report pathways and results you describe at the end of section 3.3.
Response: The newly uploaded supplementary material Table 6 is about data for clusters KEGG enrichment, and supplementary Table 6 is where I selected related genes that I describe at the end of section 3.3.
Comments 10: I suggest to provide a supplementary file reporting information of HSPs described and reported in figure 3 (similar to supplementary table 5 fot TFs).
Response: The newly uploaded supplementary material Table 7 is about the HSPs described.
Comments 11: Discussion in some points need to be better linked with the obtained results. The literature cited is informative and the mechanisms of stress response are well described and discussed, but these information need to be related with the results you obtained. The question is: are the results obtained by this experiment in line with the mechanisms already known and described?
Response: Thanks for your advice. The discussion part has been partially modified. In this part, I mainly want to discuss the related pathway and HSPs, indicating its importance in heat stress response of Camellia japonica.

Reviewer 2 Report
Comments and Suggestions for Authors
The paper mainly showed that the changes in the physiological indices and molecular networks of two C. japonica varieties under heat stress. The results were very interesting with comparing both of a resistant and a sensitive varieties. But, some of questions and revision points are exist as follows.
*Relationship or connectivity between Results and Discussion parts
-This paper are very weak at connection of story between results in this study and discussion. Except phenotype and physiological indices, discussion part is not fully described (p. 14 ~ p. 15).
*Detail explanation of the title of Figures 3 to Figures 8.
-The title of Table of Figures must provide enough information. also, the title of Table of Figures must be explain independent (This mean plentiful information should be include in the Table of Figures parts).
*Names of all DNAs must be shown Italic characters, and those of Protein’s be shown normal characters.
*Other minor revision is checked on the manuscript paper.

-Minor editing of English language required.
Author Response
Comments 1: This paper are very weak at connection of story between results in this study and discussion. Except phenotype and physiological indices, discussion part is not fully described (p. 14 ~ p. 15).
Response: Thanks for your advice, the discussion section has been partially modified. In this part, I mainly want to discuss the related pathway and HSPs, indicating its importance in heat stress response of Camellia japonica.
Comments 2: The title of Table of Figures must provide enough information. also, the title of Table of Figures must be explain independent (This mean plentiful information should be include in the Table of Figures parts).
Response: Thanks for your correction, the title of Table and Figures has been revised.
Comments 3: Names of all DNAs must be shown Italic characters, and those of Protein’s be shown normal characters.
Response: Thanks for your correction; related names have been revised.
Comment 4: Other minor revision is checked on the manuscript paper.
Response: The information you marked has been revised.

Reviewer 3 Report
Comments and Suggestions for Authors
The article have a good potential but need to improve methodology and its description, must add several citations.
As there is no line count, i will mention section, paragraph and line ..
1. [Introduction - Third paragraph - line 7] Please use the full name for MDA as it is the first time mentioned.
2. [Introduction - Fourth paragraph - lines 1 to 6] Please improve the redaction, it is a bit confusing.
3. [2.2 physiological analysis - third paragraph - line 3] Please use the full name for PBS, not mentioned before.
4. [2.2 physiological analysis - fifth paragraph - lines 5 - 8] Can you add a citation or mention where the parameters 8.02 and 20.21 come from?
5. [2.2 physiological analysis - sixth paragraph ] Here you mention the statistical analysis, but no method was mentioned, please describe the statistical tests used.
6. [2.3 Transcriptomics ] This section lack of information, you should start mentioning how many libraries you sequenced (18 it seems) and the platform used (... on an Illumina Novaseq 6000 platform for 150 cycles paired-end ...?).
Here you mention clean reads, but there is no description of the softwares used to clean the reads (and its version).
Please add the software version for Trinity, Corset, DIAMOND, HMMER, etc used on this section.
Here is no information about the evaluation of the transcriptome assembled, did you run BUSCO to assess completeness? have you tried another assembler, Trans-Abyss sometimes perform better than trinity in term of duplicated sequences.
There is no information about what gene/transcript abundace quantification method was used.
Why did you used a Log2FC of 1 as threshold? two fold change is not so strict and you will have a lot of significant genes.
Here you should mention the clustering (k-means) methodology that you show later in results. Also, probably trying to group your several thousands of DEG in six groups wont give much information, I would recommend an approach like Self - Organizing Maps to do the clustering, check kohonen package and how it is used for transcriptomics.
Also, check BioNERO package for network construction and comparison, you have two contrasting genotypes, it may help to identify important genes.
7. [2.4 Quantitative Real-Time PCR ]
Have you validated the stability of your reference gene?
8. [3.2 Global transcriptome changes]
It is more informative talk about the total number of reads sequenced and its average per sample (20 million + are recommended for transcriptomics)
The amount of transcripts is too high (There is no genome with 300K genes), you will probably have too much noisy sequences, I recommend check Trans-Abyss assembler.
9. [3.3 Time-course RNA-seq analysis - First paragraph]
This paragraph is confusing, from your PCA it can be observe than two cultivars have different expression profiles, by contrasting them you will have a lot of noise from DEG because of their genetics. I would recomend doind DEG among time points, 00 vs 16 and 16 vs 32, to obtain genes that vary across your experimental condition and then compare the groups of DEG between cultivars using SOM or the network approach mentioned before. That would reduce greatly your candidates, 60k is too much.
10. [3.3 Time-course RNA-seq analysis - Second paragraph]
Here you present results about enrichment analysis, but you have not mentioned it before in methods, what package did you used?
The figures 4 to 8, containing the whole KEGG maps, should be reduced to highlight your section of interest, or be reduced to a table. I takes too much space and are not very informative, did you contrasts cultivars?
As I mentioned before, the article have interesting results, but the methodology should be improved.
Author Response
Comments 1: [Introduction - Third paragraph - line 7] Please use the full name for MDA as it is the first time mentioned.
Response: The full name of MDA has been added.
Comments 2: [Introduction - Fourth paragraph - lines 1 to 6] Please improve the redaction, it is a bit confusing.
Response: This section has been revised, thanks for you correction.
Comments 3: [2.2 physiological analysis - third paragraph - line 3] Please use the full name for PBS, not mentioned before.
Response: The full name of PBS has been added.
Comments 4: [2.2 physiological analysis - fifth paragraph - lines 5 - 8] Can you add a citation or mention where the parameters 8.02 and 20.21 come from?
Response: The parameters come from reference 28.
Comments 5: [2.2 physiological analysis - sixth paragraph ] Here you mention the statistical analysis, but no method was mentioned, please describe the statistical tests used.
Response: The statiscal tests has been added in section 2.2.
Comments 6: [2.3 Transcriptomics ] This section lack of information, you should start mentioning how many libraries you sequenced (18 it seems) and the platform used (... on an Illumina Novaseq 6000 platform for 150 cycles paired-end ...?).
Here you mention clean reads, but there is no description of the softwares used to clean the reads (and its version).
Please add the software version for Trinity, Corset, DIAMOND, HMMER, etc used on this section.
Here is no information about the evaluation of the transcriptome assembled, did you run BUSCO to assess completeness? have you tried another assembler, Trans-Abyss sometimes perform better than trinity in term of duplicated sequences.
There is no information about what gene/transcript abundace quantification method was used.
Why did you used a Log2FC of 1 as threshold? two fold change is not so strict and you will have a lot of significant genes.
Here you should mention the clustering (k-means) methodology that you show later in results. Also, probably trying to group your several thousands of DEG in six groups wont give much information, I would recommend an approach like Self - Organizing Maps to do the clustering, check kohonen package and how it is used for transcriptomics.
Also, check BioNERO package for network construction and comparison, you have two contrasting genotypes, it may help to identify important genes.
Response: Related methods and information have been added in Section 2.3. We also noticed that transcriptome data may not be good enough, but we found the information we wanted through transcriptome. Thank you for your suggestion, and we will improve the transcriptome research method in future research.
Comments 7: Have you validated the stability of your reference gene?
Response: We selected the reference gene from database, and have validated the stability of this reference gene.
Comments 8: It is more informative talk about the total number of reads sequenced and its average per sample (20 million + are recommended for transcriptomics)
The amount of transcripts is too high (There is no genome with 300K genes), you will probably have too much noisy sequences, I recommend check Trans-Abyss assembler.
Response: Thank you for your suggestion, and we will improve the transcriptome research method in future research.
Comments 9: This paragraph is confusing, from your PCA it can be observe than two cultivars have different expression profiles, by contrasting them you will have a lot of noise from DEG because of their genetics. I would recomend doind DEG among time points, 00 vs 16 and 16 vs 32, to obtain genes that vary across your experimental condition and then compare the groups of DEG between cultivars using SOM or the network approach mentioned before. That would reduce greatly your candidates, 60k is too much.
Response: Thanks for your advice, contrasting them do have a lot of noise, for more comprehensive information, we have two contrast method, contrast two cultivars and 00 vs16 and 00 vs 32 in one cultivars.
Comments 10: Here you present results about enrichment analysis, but you have not mentioned it before in methods, what package did you used?
The figures 4 to 8, containing the whole KEGG maps, should be reduced to highlight your section of interest, or be reduced to a table. I takes too much space and are not very informative, did you contrasts cultivars?
Response: The methods of enrichment analysis has been added in section 2.3. We think the pathways of figures 4 to 8 are imporatant in heat stress response of Camellia japonica, these pathways may be future research directions. We have contrasts cultivars, and we found three pathways may be the reason of different tolerance.
Round 2
Reviewer 2 Report
Comments and Suggestions for Authors
Most of revision points were revised by authors.
My question is; Description of the meaning/or relation between results of this study and discussion is needed at discussion part. Authors described plentiful information about HSPs and TFs at discussion part (p.15 to p. 16), however, description of the relation/or meaning point between discussion and the results of the screened HSPs and TFs ( p. 19, p.14), is not found.

Minor editing of English language required.
Author Response
Comments 1:My question is; Description of the meaning/or relation between results of this study and discussion is needed at discussion part. Authors described plentiful information about HSPs and TFs at discussion part (p.15 to p. 16), however, description of the relation/or meaning point between discussion and the results of the screened HSPs and TFs ( p. 19, p.14), is not found.
Answer: Thanks for your correction. Relevant descriptions have been added to the disscuison section, and english language has been edited.

Reviewer 3 Report
Comments and Suggestions for Authors
The authors have resolved most of the issues previously mentioned, and the article clearity has been improved.
Author Response
Comments 1: The authors have resolved most of the issues previously mentioned, and the article clearity has been improved.
Answer: Thanks for your suggestion and support!